# INTERACTIVELY PROVIDING EXPLANATIONS FOR TRANSFORMER LANGUAGE MODELS

## ABSTRACT

Transformer language models are state-of-the-art in a multitude of NLP tasks. Despite these successes, their opaqueness remains problematic. Recent methods aiming to provide interpretability and explainability to black-box models primarily focus on post hoc explanations of (sometimes spurious) input-output correlations. Instead, we emphasize using prototype networks directly incorporated into the model architecture and hence explain the reasoning process behind the network's decisions. Moreover, while our architecture performs on par with several language models, it enables one to learn from user interactions. This not only offers a better understanding of language models but uses human capabilities to incorporate knowledge outside of the rigid range of purely data-driven approaches.

## 1 INTRODUCTION

Transformer language models (LMs) are ubiquitous in NLP today but also notoriously opaque. Therefore, it is not surprising that a growing body of work aims to interpret them: Recent evaluations of approaches of saliency methods (Ding & Koehn, 2021) and instance attribution methods Pezeshkpour et al. (2021) find that, while intriguing, for the same outputs, different methods assign importance to different inputs. Furthermore, they are usually employed post hoc, thus possibly encouraging reporting bias (Gordon & Van Durme, 2013). The black-box character of LMs becomes especially problematic, as the data to train them might be unfiltered and contain (human) bias. As a result, ethical concerns about these models arise, which can have a substantial negative impact on society as they get increasingly integrated into our lives (Bender et al., 2021).

Here, we focus on providing case-based reasoning explanations during the inference process, directly outputting the LM's predictions. In doing so, we avoid the problems mentioned above of post hoc explanations and help to reduce the issue of (human) bias. To increase the interpretability of the model, we enhance the transformer architecture with a prototype layer and propose *Prototypical-Transformer Explanation* (Proto-Trex) Networks. Proto-Trex networks provide an explanation as a prototypical example for a specific model prediction, which is similar to (training-)samples with the corresponding label.

Our experimental results demonstrate that Proto-Trex networks perform on par with non-interpretable baselines, *e.g.* BERT (Devlin et al., 2019) and GPT (Radford et al., 2019). In terms of explanations, we show promising results with learned prototypes providing helpful explanations for the user to understand better the LMs decision-making, which, in turn, increases trust and reliability. To further enhance the prototypical network with human supervision, we propose an interactive learning setting (iProto-Trex) that allows users of any knowledge to give feedback and improve the model (Ribeiro et al., 2016), moving beyond a purely data-driven approach.

To summarize, our contributions are as follows: We (i) introduce prototype networks for transformer LMs that provide explanations and (ii) show that they are on par with non-interpretable baselines on classification tasks on different architectures. Furthermore, to improve prototype networks, we (iii) provide a novel interactive prototype learning setup accounting for user feedback certainty.

We proceed as follows. We start by briefly reviewing related work of interpretability in NLP. Then we introduce Proto-Trex networks, including our novel interactive learning setup combining explanatory interactive learning with prototype networks. Before concluding, we discuss faithful explanations and touch upon the results of our experimental evaluation.

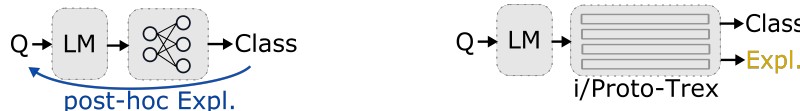

| Explainer | Q: Staff is available to assist, but limited to the knowledge and understanding of the individual. |
|---|---|
| Post hoc | Staff is available to assist, but limited to the knowledge and understanding of the individual. |
| Proto-Trex | They literally treat you like you are bothering them. No customer service skills. |
| iProto-Trex | They offer a bad service. |

Figure 1: i/Proto-Trex compared to a post hoc explanation on sentiment classification. The input query (top row) is classified (as negative), and three explanations are provided. The first (post hoc) explanation is provided by LIT (Tenney et al., 2020) with LIME (Ribeiro et al., 2016). The intensity of the color denotes the influence of a certain word: Blue (red) color indicates positive (negative) sentiment. Middle and bottom row explanations are provided by (interactive) Proto-Trex networks.

## 2 TOWARDS THE EXPLAINABILITY OF TRANSFORMER LANGUAGE MODELS

To open the black-box of transformers, we use i/Proto-Trex networks built upon post hoc interpretation methods, case-based reasoning approaches, and explanatory interactive learning.

**Post hoc interpretability.** Various (post hoc) interpretability methods focus on different parts of the transformer architecture. Atanasova et al. (2020) and Belinkov & Glass (2019) provide overviews of this fast developing field. Generally, there are methods that analyze word representations (Voita et al., 2019a), the attention distribution throughout the model (Jain & Wallace, 2019; Wiegreffe & Pinter, 2019), and the (attention and classification) model heads (Voita et al., 2019b; Geva et al., 2021). Other approaches such as Geva et al. (2020) focus on the feed-forward layers. Gradient-based approaches, such as Sundararajan et al. (2017) and Smilkov et al. (2017) can generally be used to trace gradients, while influence functions (Koh & Liang, 2017) trace model parameter changes throughout a LM (Han et al., 2020). While backtracking all the model weights might be possible, such explanations can only tell us which part of the input they are looking at (Chen et al., 2019) and humans are ill-equipped to interpret them (Stammer et al., 2021). So instead, we aim for more intuitive and sparse explanations: well-descriptive but short sequences as prototypes. These are not only able to reveal the parts of the input it is looking at but also show prototypical cases similar to those parts.

**Case-based Reasoning in Deep Neural Networks.** Our work relates most closely to previous work demonstrating the benefits of prototype networks in Computer Vision (Li et al., 2018; Chen et al., 2019) as well as for sequential data by combining them with RNNs (Hase et al., 2019; Ming et al., 2019). More precisely, in contrast to post hoc interpretation methods, prototype networks employ explanations in the, *e.g.* classification process, *i.e.* classify a sample by comparing several parts – which could be words of a sentence– to (learned) prototypical parts from other samples for a given class. If things *look* similar, they are classified similarly. Humans behave very similarly, which is called case-based reasoning (Clifton & Frohnsdorff, 2001). The prototypical parts provided by the network help the user to understand the classification. Tab. 1 illustrates the benefit of prototypical explanations. One can observe that the post hoc explanation can leave the user clueless, while the prototypical explanations can help better understand the network's decision. Accordingly, prototypes are an additional method in the interpretability toolbox, extending current post hoc methods. As described, the reasoning process of prototypical networks is qualitatively similar to that of humans (Clifton & Frohnsdorff, 2001; Chen et al., 2019). In previous approaches, however, the training of such networks was entirely data-driven. Therefore, like Chen et al. (2019), in our work, we do not focus on quantifying unit interpretability of prototypical networks, but extend learning the reasoning process of our network by Explanatory Interactive Learning (Schramowski et al., 2020).

**Explanatory Interactive Learning** (XIL) incorporates explanatory supervision in the learning process by involving the human user. More precisely, human users can ask the model to explain a prediction. They can then respond by correcting the model if necessary, providing a slightly improved –but not necessarily optimal– feedback on the explanations (Stammer et al., 2021). Interactive learning and, in particular, explanatory interactive learning showed not only to increase the explanation quality of deep black-box models but also the trust of the user (Ross et al., 2017; Selvaraju et al.,

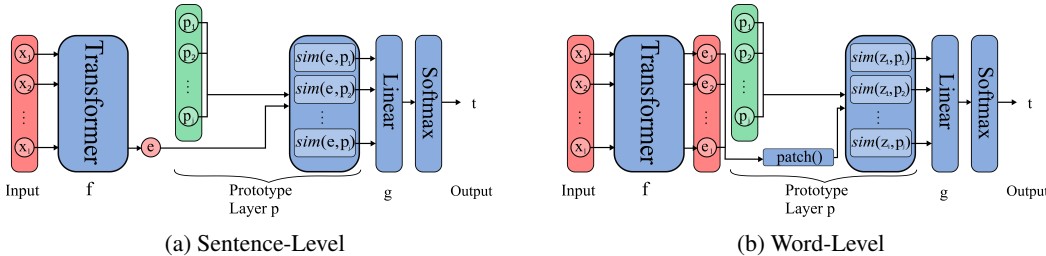

Figure 2: Architecture of the Prototypical-Transformer Explanation Network (Proto-Trex): (a) Sentence-level case uses a transformer to compute sentence-level embedding explanations, while (b) word-level case provides multiple word-level explanations and therefore requires an additional word-selection layer.

2019; Teso & Kersting, 2019; Schramowski et al., 2020). In contrast to previous (post hoc) XIL methods, we avoid tracing gradients. That makes our approach particularly efficient, as we only change prototypes (*cf.* Fig 4).

## 3 PROTO-TREX: PROTOTYPE LEARNING FOR TRANSFORMER LMS

The prototype network (Proto-Trex) architecture builds upon large-scale transformer LMs, summarized in Fig. 2. In the following, we describe the general idea of prototypical networks and our transformer-based prototype architecture and its modules in more detail.

### 3.1 PROTO-TREX NETWORKS

**General Idea.** Proto-Trex is a prototype network, where prototypes are representative observations in the training set. The classification is based on the neighboring prototypes, *i.e.* an input query is classified as positive when the network thinks it looks similar to a certain (near) positive prototype within the training set. In more detail, the reasoning and classification processes use the same module, *i.e.* , they use the similarity between query and prototypes. As both reasoning and classification use the same module (in parallel), we consequently have an interpretable model. In particular, (interpretable) prototypical explanations help verify Right for the Right Reasons (Ross et al., 2017).

**Architecture.** Specifically, the model, in our case the pre-trained transformer $f$, creates a context embedding $\mathbf{e}$ from the input sequence $\mathbf{x}$. This embedding contains useful features for the prediction and is then passed on to the prototype layer. This layer computes the similarity between the embedding $\mathbf{e}$ and each of the trainable prototypes $\mathbf{p}_j$ from the set of prototypes $\mathcal{P}$. The prototypes are learned during the training process and represent prototypical (sub-)sequences in the training data. We use transformer models both on the sentence-level (a) and the word-level (b), resulting in a single input representation for the sentence-level transformer and $l$ for word-level. To compare the word-level embeddings $\mathbf{e}_i$ with the prototypes, we have to patch them ($\mathcal{Z} = \{patch(\mathbf{e}_i)\}$) into subsequences according to the desired prototype length $k$. The resulting similarity values are passed on to the final linear layer $g$ with weight matrix $\mathbf{w}_g$ and no bias, mapping from $m$ (the number of prototypes) to the number of classes. Finally, the output values are passed on to a softmax producing class probabilities and predictions $t$.

### 3.2 PROTO-TREX LOSS

Optimization of prototype networks aims at maximizing both performance and interpretability. To this end, our Proto-Trex loss $\mathcal{L}$ combines performance and interpretability with prior knowledge of the prototype network's structure. Let us illustrate this for the word-level case (the sentence-level case results from $\mathbf{e} = \mathbf{z}$): $\mathcal{L} :=$

$$\min_{\mathcal{P}, \mathbf{w}_g} \frac{1}{n} \sum_{i=1}^{n} \text{CE}(t_i, y_i) + \lambda_1 \text{Clst}(\mathbf{z}, \mathbf{p}) + \lambda_2 \text{Sep}(\mathbf{z}, \mathbf{p}) + \lambda_3 \text{Distr}(\mathbf{z}, \mathbf{p}) + \lambda_4 \text{Divers}(\mathbf{p}) + \lambda_5 ||\mathbf{w}_g||$$

$$(1)$$

where $\lambda_i$ weights the influence of the different terms and $n$ is the number of training examples. The first term is the cross-entropy (CE) loss optimizing the predictive power of the network. The second term (Clst) clusters the prototypes w.r.t. the training examples of the same class, maximizing similarity to them (we rewrite maximization as minimization terms), and the separation loss minimizes the similarity to other-class instances:

$$\text{Clst}(\mathbf{z}, \mathbf{p}) := -\frac{1}{n} \sum_{i=1}^{n} \min_{j:\mathbf{p}_j \in \mathcal{P}_{y_i}} \min_{\mathbf{z} \in \mathcal{Z}} \text{sim}(\mathbf{z}, \mathbf{p}_j) \; ; \quad \text{Sep}(\mathbf{z}, \mathbf{p}) := \frac{1}{n} \sum_{i=1}^{n} \min_{j:\mathbf{p}_j \notin \mathcal{P}_{y_i}} \min_{\mathbf{z} \in \mathcal{Z}} \text{sim}(\mathbf{z}, \mathbf{p}_j) \quad (2)$$

Together, Clst and Sep push each prototype to focus more on training examples from the same class and less on training examples from other classes. Both are motivated by ProtoPNet (Chen et al., 2019). To get prototypes that are distributed well in the embedding space, we introduce two additional losses, a distribution loss (Li et al., 2018), assuring that a prototype is nearby each training example, and a diversity loss (Ming et al., 2019):

$$\text{Distr}(\mathbf{z}, \mathbf{p}) := -\frac{1}{m} \sum_{j=1}^{m} \min_{i \in [1,n]} \min_{\mathbf{z} \in \mathcal{Z}} \text{sim}(\mathbf{z}, \mathbf{p}_j) \; ; \quad \text{Divers}(\mathbf{p}) := \frac{1}{m} \sum_{\hat{j}=1}^{m} \min_{j:\mathbf{p}_j \in \mathcal{P}} \text{sim}(\mathbf{p}_{\hat{j}}, \mathbf{p}_j) \quad (3)$$

In contrast to the other terms, the diversity loss does not compute similarities between embeddings and prototypes but between prototypes themselves. It is another way of distributing prototypes in the embedding space as it maximizes the distance between prototypes, preventing them from staying at the same –not necessarily optimal– location. This is especially helpful in the case of multiple prototypes for a single class, encouraging them to represent different facets of the class. If they are otherwise too similar, no information is gained and resulting in redundant prototypes – together with the class-specific loss encouraged by the cluster loss (Clst) and the separation loss (Sep), this helps to compute prototypes that focus solely on their class. Otherwise, we can get ambiguous prototypes leading to negative reasoning. Also, we clamp the weights of the classification layer with $\min(\mathbf{w}_g, 0)$ to avoid negative reasoning (Chen et al., 2019). Finally, the last term of the Proto-Trex loss (Eq. 1) is an L1-regularization term of the last layer ($g$), which prevents the network from overfitting or relying too much on a single prototype.

### 3.3 SIMILARITY COMPUTATION

Computing similarities is an essential aspect of Proto-Trex. For prototypes to represent certain aspects or features of the input distribution in the embedding space, we compute the similarity $\text{sim}(\mathbf{e}, \mathbf{p})$ between an embedded training example and a prototype. A distance minimization can replace each similarity maximization $\max \text{sim}(\mathbf{e}, \mathbf{p}) = \min \text{dist}(\mathbf{e}, \mathbf{p})$. We here follow common practice and explore both the L2-norm or the cosine similarity:

$$\text{sim}(\mathbf{e}, \mathbf{p}_j) = \begin{cases} -\|\mathbf{e} - \mathbf{p}_j\|_2 & \text{, L2-norm} \\ \frac{\mathbf{e} \cdot \mathbf{p}_j}{\|\mathbf{e}\|_2 \|\mathbf{p}_j\|_2} & \text{, cosine similarity} \end{cases} \, ,$$

where the index $j$ denotes a specific prototype. For each training example, there is an embedding $\mathbf{e}$ which is compared to all $m$ prototypes $\mathbf{p}$, *i.e.* we get $m$ similarity values for each embedding.

The L2-norm assumes a Gaussian prior, which can be a wrong assumption, which is why we also investigate cosine similarity. While L2-norm computes the distance between two vectors, cosine similarity measures the angle between them. Both, but especially cosine similarity, are natural choices for NLP tasks (Manning et al., 2008). However, cosine similarity is more robust here than L2-norm as the distance's magnitude between vectors has no influence due to normalization.

### 3.4 SELECTION FOR WORD-LEVEL PROTOTYPES

Since learning prototypes for LMs pre-trained on word-level representations is more involved than for the sentence-level, let us focus on them here; the sentence-level case naturally follows from the discussion. Word-level prototypes are generally sensible as explanations should consist of sparse sequences, at best focusing only on subsequences of the input sentence. As sequences can also be ambiguous and contain little information, we combine different word embeddings ($patch()$) and enforce the prototype to be similar to the relevant subsequences, containing the most information. But how do we select the most informative words?

(a) Sliding windows approach: Convolutions sliding over a sequence, (1) without and (2) with additional dilation in the convolutional window.

(b) Word selection with self-attention. First, attention weights are calculated for each word, and most attended-to words are selected; then all patches w.r.t. prototype length are calculated.

Figure 3: Word selection for word-level Proto-Trex with (a) convolution and (b) attention.

A naive approach would be to simply compute all possible patches (word combinations) of the input sequence and compare them to the prototypes to find the best patch for classification – the most important subsequence should then be similar to a certain prototype. Unfortunately, for long input sequences with length $l$ it becomes hard to compute all possible word patches of length $k$, in this case $|\mathcal{Z}| = \binom{l}{k}$. To solve this problem without losing too many valuable patches, the following two approaches are sensible ideas.

**(a) Sliding windows** naturally reduce the number of patches. A sliding window is a convolution that selects a certain part of the input according to the window size and then sliding to the next part. The main disadvantage of sliding windows is the relatively rigid structure of a window. This is problematic in the context of NLP tasks that often have long-range dependencies. We introduce dilation to loosen this. Dilation facilitates looking at direct word neighborhoods but also at more distant ones to capture long-range dependencies. This "convolutional" approach is illustrated in Fig. 3(a). We note that applied to the contextualized word embeddings of transformer LMs, the convolutional approach without dilation should contain not only local information but also some global information. Unfortunately, this information is stored in the embeddings and cannot be visualized easily.

As an alternative, the **(b) Self-Attention** approach adds a self-attention layer after the transformer $f$ and before the prototype layer $p$ to filter irrelevant words (with low attention-scores), *cf.* Fig. 3(b). The purpose of this self-attention layer differs from the one in the transformer LM itself. The here used attention mechanism selects the most important words, and the embedding representation remains untouched. To still provide as much information and variety as possible, the number of selected words $n_w$ of the attention layer which are passed into the distance computation is a hyperparameter, and we chose it to be twice as much as the length of a single prototype however clamped by the threshold $k_{\lim}$: $n_w = min(max(k_{\lim}, k), 2k)$. The threshold can be set w.r.t. computational efficiency. This means having a prototype of length $k = 4$ and $k_{\lim} = 10$, the attention layer selects $n_w = 8$ words yielding in this example $|\mathcal{Z}| = \binom{8}{4} = 70$ patches for the distance computation.

## 3.5 Decoding via Nearest Neighbor Projection

Proto-Trex networks encode prototypes in the embedding space. Consequently, they cannot simply be decoded from the transformer embedding space, as this space and textual data are categorical and not continuous. To overcome this, we assign, *i.e.* project, each prototype to its nearest neighbor in the training data (where the textual representation is available) in an intermediate training step. Thereby, we ensure that the prototype really represents what it actually looks like, which also increases the interpretability of the Proto-Trex network. The final training step (after the nearest neighbor projection) then fine-tunes the classification head to adapt it to the projected prototypes. Doing so has the advantage of being decoded precisely on the spot and providing more certainty for the explanatory power. The disadvantage, however, is that a prototype may be located sub-optimally, which could lead to performance losses. We faced this issue especially for word-level prototypes and found the projection ineligible for these highly contextualized word representations. Instead, for word-level, we only use the nearest neighbor for approximating the prototype to provide the explanation.

## 4 Interactive Prototype Learning

Recent case-based reasoning approaches in Computer Vision and NLP usually assume a large volume of data for training (Li et al., 2018; Chen et al., 2019; Hase et al., 2019), with little or no

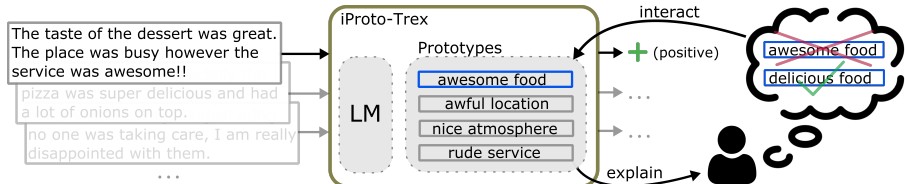

Figure 4: Interactive Prototype Learning: iProto-Trex classifies the input and gives the user an explanation based on a prototype. The explanation is highlighted in bright blue. If the user is dissatisfied with the given explanation, they can replace it with a self-chosen sequence.

user feedback during the model building process. However, user knowledge and interaction, in particular via explanations, can be valuable already in the model building and understanding phase, significantly reducing the amount of data required, avoiding Clever-Hans moments early on, and increasing the explanation quality of the model and, in turn, the trust of the user, see e.g. Teso & Kersting (2019); Schramowski et al. (2020).

**Interactive Proto-Trex.** To address this, we propose an explanatory interactive learning approach for Proto-Trex networks, called iProto-Trex, as illustrated in Fig. 4. Specifically, carried out during the training progress by freezing and evaluating the current state or post hoc training, the interaction takes the following form. At each step, the Proto-Trex networks provide prototypes as explanations of a classification. The user responds by correcting the learner, if necessary, in our case, the prototypes. For this, iProto-Trex provides a range of opportunities for interacting with prototypes.

Specifically, iProto-Trex distinguishes between weak-knowledge and strong-knowledge interactions. In strong-knowledge interactions, users are certain about their feedback (which could require high-level expertise). In this case, iProto-Trex provides options to *remove* and *add* prototypes. Removing a prototype is also helpful if the network has learned redundant prototypical sequences or prototypes that cover all critical aspects of the task at hand. If users are content with all the present prototypical sequences, they can also add new prototypes (*Replacing* a prototype is the same as removing plus adding a prototype). After these interactions, the subsequent classification layer is retrained to integrate and balance the new prototype setup in its decision-making.

In weak-knowledge interactions, users only state content or discontent with a prototype based on their intuition. They do not need to know what a replacement should exactly look like, which is a good trade-off between user knowledge and loss optimization of the network. To this end, iProto-Trex uses *re-initialization* and *fine-tuning*; instead of providing an explicit replacement, users freeze prototypes they like while relearning those they are dissatisfied with. The difference between these approaches is that users can express how well the current prototype represents a particular task aspect. Another form of weak-knowledge interaction, which iProto-Trex provides only for sentence-level prototypes, is *pruning* the sequence length of prototypical explanations, *i.e.*, compressing it to the essentials. In order to limit meaning changes of a sequence, a threshold is provided for the cosine similarity of the pruned and the original version. We initially set this threshold to $0.8$, but users can set the threshold as they like, resulting in a strong-knowledge interaction.

**Soft User Feedback.** iProto-Trex's interaction methods, especially the strong-knowledge ones, require users to be quite confident about their feedback –they have to be experts. To elevate this burden, we propose a *soft feedback* mechanism, using a loss based on the prototype similarity as

$$\mathcal{L}_{\text{interact}} := \lambda_6 \max_{\mathbf{p}_{\text{old}}} \big( \max(\text{sim}(\mathbf{p}_{\text{old}}, \mathbf{p}_{\text{new}}), c) \big) \qquad (4)$$

with $c \in [0, 1]$. Instead of directly replacing the selected prototype $\mathbf{p}_{\text{old}}$, this soft interaction loss pushes the prototypes to be similar to the desired prototype $\mathbf{p}_{\text{new}}$ as suggested by a user. Moreover, users can control how strong iProto-Trex should incorporate their interventions by setting a certainty value $c$ between $0$ (low certainty) and $1$ (high certainty).

## 5 FAITHFULNESS OF PROTOTYPICAL EXPLANATIONS

A question that naturally arises when dealing with explainable AI is how to evaluate the quality of an explanation. One of the key metrics in NLP and Computer Vision is the faithfulness of explanations

(Jacovi & Goldberg, 2021; DeYoung et al., 2020), *i.e.* whether a given explanation represents the true reasoning process of the black-box model. We apply and adjust this metric to text-based prototype networks. First, we evaluate the faithfulness of Proto-Trex as a whole (DeYoung et al., 2020) by showing that the class probabilities change significantly if we perturb the input. We follow their approach of computing comprehensiveness and sufficiency. To calculate comprehensiveness, we remove the rationale in each input sequence and evaluate the changes in the class probabilities for the prediction. Second, we perturb the prototype layer by removing the top explanation for a given test sample and compute the loss in accuracy. This helps to identify to what extent the decision was actually based on the explanation given. Since our prototypical explanations are already short sequences, removing the rationale in the explanation is similar to removing the entire prototype.

# 6 EXPERIMENTAL EVALUATION

Our intention here is to investigate how good prototypes help to understand transformer LMs. To this end, we evaluated i/Proto-Trex explanations on three benchmark datasets: MovieReview (Pang et al., 2002), Open Yelp[1] and Jigsaw Toxicity[2]. We compared five pre-trained LMs (GPT-2 (Radford et al., 2019), BERT (Devlin et al., 2019), DistilBERT, (Sanh et al., 2020), SBERT (Reimers & Gurevych, 2019) and the text-encoder of CLIP (Radford et al., 2021)) to investigate three questions:

**(Q1)** How much does adding a prototype layer affect the performance of (non-interpretable) LMs, *i.e.* , a classification head defined by two fully-connected non-linear layers?

**(Q2)** How does the performance change after and during interaction with the model explanations? In particular, we investigated the different modules of Proto-Trex networks on sentence- and word-level and the interaction between users and the prototypical explanations.

**(Q3)** How faithful are the given explanations, *i.e.* how well does the given explanation represent the true reasoning of the black-box model?

We present qualitative and quantitative results and refer to the Appendix for additional details on the experiments and our implementation, as well as additional qualitative results. If not stated otherwise, the Proto-Trex architecture includes sentence- and word-level embeddings, the convolution module without dilation to select word-tokens, and cosine similarity to compute the similarity between prototypical explanations and input query. We optimized the prototype and classification module with the proposed loss (Eq. 1) and initialize the prototypes randomly. In each experiment, the number of prototypes is $m = 10$, and the largest variant of the corresponding pre-trained LM is evaluated.

**(Q1,2) Trade off accuracy and interpretability.** Tab. 1 summarizes the experimental result of Proto-Trex network based on different sentence- and word-level LMs. As one can see and expected from the literature, interpretability comes along with a trade-off in accuracy. The trade-off is generally higher on sentence-level LMs, partially due to the nearest neighbor projection (*cf.* Appendix for direct comparison). However, the difference between traditional LMs and Proto-Trex LMs is often marginal and task-dependent (*e.g.* DistilBERT on Yelp and Movie). Surprisingly, in the case of BERT and GPT-2 (on Yelp and Toxicity), the Proto-Trex network is outperforming the baseline LMs. Most interestingly, one can see that the user may boost the performance of the corresponding Proto-Trex network interactively (iProto-Trex). Overall, i/Proto-Trex is competitive with state-of-the-art LMs while being much more transparent.

**(Q1) Ablation study.** Next, we investigate different Proto-Trex module choices for performance, *cf.* Tab. 1b), and explanation provision impact. While input- and prototype similarity computation does not explicitly correlate with the prototypes learned –yet we assumed it to be the better choice–, the word-selection module also impacts the explanation outcome. Furthermore, cosine similarity is not only more accurate but also converges faster. In terms of accuracy, the convolutional word selection outperforms the attention module. More importantly, we also observe an advantage for the provided explanations; namely, the attention module tends to select punctuation and stop words. According to Ethayarajh (2019), this is because punctuation and stop-words are among the most context-specific word representations–they are not polysemous themselves but can have an infinite number of contexts. In contrast, the convolution module is easier to interpret as words are coherent.

---

[1]https://www.yelp.com/dataset
[2]https://www.kaggle.com/c/jigsaw-toxic-comment-classification-challenge

**a)**

| | Language Model | Yelp | Movie | Toxicity |
|---|---|---|---|---|
| sent.-level | SBERT | 94.92 | 84.56 | ∘**84.40** |
| | SBERT (Proto-Trex) | 93.59 | 80.05 | 73.19 |
| | CLIP | 93.78 | 75.49 | 80.82 |
| | CLIP (Proto-Trex) | 87.16 | 63.52 | 67.75 |
| word-level | BERT | 89.41 | 61.45 | 76.88 |
| | BERT (Proto-Trex) | 92.10 | 75.51 | 79.35 |
| | GPT-2 | 93.78 | ●**87.05** | 81.56 |
| | GPT-2 (Proto-Trex) | ●**95.32** | 84.57 | 84.14 |
| | DistilBERT | 92.91 | 79.62 | 81.57 |
| | DistilBERT (Proto-Trex) | 92.71 | 78.64 | 83.39 |
| inter. | SBERT (iProto-Trex) | 93.81 | 80.24 | 73.36 |
| | GPT-2 (iProto-Trex) | ∘**95.25** | ∘**84.80** | ●**84.51** |

**b)**

| | Word-Selection | | Similarity | |
|---|---|---|---|---|
| Proto-Trex LM | Conv. | Attn. | Cos. | L2 |
| BERT | **92.10** | 89.66 | **92.10** | 91.34 |
| GPT-2 | **95.32** | 94.16 | **95.32** | 94.99 |
| DistilBERT | **92.71** | 91.09 | **92.71** | 92.05 |
| SBERT | – | – | **93.59** | 93.13 |
| CLIP | – | – | **87.16** | 86.88 |

**c)**

| | Faithfulness | | Acc. | |
|---|---|---|---|---|
| Proto-Trex LM | Comp. | Suff. | before | after |
| SBERT | 0.22 | –0.08 | 80.05 | 69.66 |
| iSBERT | 0.21 | –0.08 | 80.24 | 70.76 |
| GPT-2 | 0.12 | 0.02 | 84.57 | 49.91 |
| iGPT-2 | 0.13 | 0.02 | 84.80 | 55.74 |

Table 1: Quantitative Results. a) Average Accuracy of Proto-Trex with different LMs on different datasets. Best ("●") and runner-up ("∘") are **bold**. b) Ablation study of Proto-Trex module choices on Yelp dataset. c) Evaluation of i/Proto-Trex regarding faithfulness on MovieReview dataset. Mean values over 5 runs are reported. The confidence intervals can be found in Appendix Tab. 5

| | Importance | Query: I really like the good food and kind service here. |
|---|---|---|
| sentence-level | 0.52·8.07= **4.20** | *P8*: Oliver Rocks! Great hidden gem in the middle of Mandalay! Great friends great times |
| | 0.84·3.04= **2.55** | *P4*: Incredibly delicious!!! Service was great and food was awesome. Will definitely come back! |
| | 0.90·1.21= **1.10** | *P6*: I found this place very inviting and welcoming. The food is great so as the servers.. love the food and the service is phenomenal!! Well done everyone..! |
| | 0.79·1.16= **0.92** | *P2*: This place is really high quality and the service is amazing and awesome! Jayden was very helpful and was prompt and attentive. Will come back as the quality is so good and the service made it! |
| interaction | 0.84·5.02= **4.22** | *P4*: Incredibly delicious!!! Service was great and food was awesome. Will definitely come back! |
| | 0.92·2.10= **1.94** | *P6*: I found this place very inviting and welcoming. The food is great so as the servers. |
| | 0.76·2.13= **1.62** | *P2*: This place is really high quality and the service is amazing and awesome! |
| | 0.73·0.71= **0.52** | *P10*: Fun and friendly atmosphere, fantastic selection. The sushi is so fresh |

Table 2: Provided Explanations by Proto-Trex networks. Top-four explanations for the query (top row) are provided for sentence-level networks. Colored boxes illustrate the advantage of pruning sentence explanations (interaction). Importance scores (left, bold) are calculated with cosine similarity and classification weight.

**(Q1,2) Provided explanations.** To analyze the provided explanations, we consider the SBERT based Proto-Trex networks, *cf.* Tab. 1, trained on the Open Yelp dataset. Tab. 2 shows how Proto-Trex provides users with explanations. These explanations are given as prototypical sequences that correspond to a query. In addition, Proto-Trex provides corresponding importance scores, indicating the significance of an explanation for the classification. We show four sentence-level prototypes for the query that users can quickly extract the important aspects that help understand the classification. However, one can observe that the sentence-level explanations are sometimes lacking sparsity and are difficult to interpret about the query, which also demonstrates the demand for (further) interaction (*cf.* colored boxes with prototypical explanations *P6* and *P2* in Tab. 2). The pruned Proto-Trex provides less ambiguous and easier to interpret sequences.

**(Q2) Interactive prototype learning.** In the previous experiment, pruning has already shown the benefits of adapting explanations to users' preferences. In order to further evaluate our interactive learning setup, we incorporated certainty ($c$) and evaluated its effectiveness. Since sentence-level

| Type of interaction | Acc. | Prototype |
|---|---|---|
| no interaction | 93.64 | Horrible customer service and service does not care about safety features. That's all I'm going to say. Oh they also don't care about their customers |
| re-initialize | 93.80 | Terrible delivery service. People are mean and don't care about their customers service. I will not ever come back to this place. Also the food is small, not very good and too expensive. |
| soft replace (0.5) | 93.81 | Terrible delivery service. People are mean and don't care about their customers service. I will not ever come back to this place. Also the food is small, not very good and too expensive. |
| soft replace (0.9) | 93.79 | I really don't recommend this place. The food is not good, service is bad. The entertainment is so cheesy. Not good |
| soft replace (1.0) | 93.79 | They offer a bad service. |

Table 3: Interactive learning: Different user interaction methods with accuracy on Yelp reviews. Interaction changes a redundant prototype into a user-selected one incorporating more user knowledge without significantly impacting the accuracy.

prototypes allow for better coherence, this is where we will focus the interactive learning. Tab. 3 shows an influential, *i.e.* high importance value, sentence-level explanation (yet no interaction) for negative restaurant reviews on the Open Yelp dataset. Assuming a user is dissatisfied with an explanation yet uncertain about what a good explanation would entail: Applying re-initialization (most uncertain interaction technique) confirms the user's intuition of a "weak" component, as the slight increase in accuracy indicates. More importantly, we can already observe that users can influence the network's decision process based on their preferences without performance loss. However, the revised explanation is still not sufficiently interpretable. Therefore, we considered incorporating explicit user feedback here. In this case, the phrase "They offer a bad service" served as soft replacement for the model's provided explanation, and the user applied it with different levels of certainty. First, a low certainty value ($c = 0.5$) results in the same explanation as before. This is because the similarity of any two prototypes with clearly negative sentiment is higher than a certainty threshold of 0.5. As the user gets more certain, he gradually increases the threshold. Finally ($c = 1$), the user simply replaced the prototype to obtain his desired solution without a trade-off in accuracy. In summary, our results demonstrate that interactive learning is a solid method to counter the network's incapability in consistently providing interpretable prototypes along with high accuracy.

**(Q3) Faithful prototypes.** To analyze faithfulness, we first follow DeYoung et al. (2020) and use the MovieReview dataset as they provide human-annotated rationales for this dataset. We compare the classification probabilities of the samples (DeYoung et al., 2020) with the ones where the rationale is removed (comprehensiveness) and with the ones with only the rationale (sufficiency). Tab. 1(c) shows that i/Proto-Trex scores high for comprehensiveness, indicating that the network is focusing on the input rationale for the classification, while low sufficiency scores indicate that the surrounding context has little to even poor impact on the prediction. This shows that i/Proto-Trex are generally faithful classifiers, *i.e.* the human-annotated rationales agree with the internal rationales in the model. Furthermore, we remove the top (prototype) explanation for the classification and examine the change in accuracy after the removal to show that it is the prototype that faithfully captures the rationale. Tab. 1(c) shows a drastic decrease in accuracy for both models after the explanation removal, confirming the faithful contribution of prototypical explanations to the classification.

## 7 CONCLUSION

Large-scale transformer LMs, like other black-box models, lack interpretability. We presented methods (prototype networks) to incorporate case-based reasoning to explain the LM's decisions. Despite the explanatory power of prototypical explanations, challenges regarding the quality of their interpretability still exist. Previous applications lack human supervision, although case-based reasoning is a human-inspired approach. Therefore, we propose an interactive prototype learning setup to overcome these challenges and improve the network's capabilities by incorporating human knowledge with the consideration of knowledge certainty. Future work covers improving methods for patching the input for word-level prototypes to enable projection for higher interpretability.

## 8 ETHICS AND REPRODUCIBILITY STATEMENT

The article presented here approaches the explainability of transformer language models through case-based reasoning in the form of prototypical explanations. The interpretable prototype module that we learned is built on a pre-trained transformer language model. As these models are trained on data that is not publicly available, it remains an open question whether any kind of bias is inherent to the model as a whole (Bender et al., 2021). In addition, concerns about privacy violations and other potential misuse emerge as these models are trained with weak to no supervision.

The code to reproduce the results can be found in our publicly available repository. Furthermore, we provide the trained Proto-Trex models to obtain explanations.

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

## A APPENDIX

### A.1 DATASETS

The benchmark datasets mentioned above provide a fixed test set except the Yelp Open dataset. For the Yelp Open dataset we randomly select $200\,000$ training examples and split them into train (70%), validation (15%) and test (15%) set. We split the Jigsaw Toxicity train set into train (20%) and validation (80%) set. Before the split, we filter out long sequences ($num\_tokens > 40$) for each dataset. This is required because transformer-based LMs can only handle sequences of limited

length, especially CLIP, and long sequences also cause the other sequences of a set to be padded to the same length, which, in turn, produces many padding tokens. For the tokenization we use the GPT2-tokenizer[3]. We apply grid search for hyperparameters optimization. We report the cross-validated results. For the faithfulness computation we use the *eraserbenchmark* repository[4] and data[5].

## A.2 TRAINING

The Proto-Trex networks are trained with and evaluated on different datasets. We use PyTorch[6] for the implementation. We optimize our model with Adam optimizer with hyperparameters $\boldsymbol{\beta} = (0.9, 0.999)$ and $\epsilon = 10^{-8}$. We use a base learning rate of $lr_{\text{base}} = 0.001$ and apply learning rate warm-up and scheduling that is here a linear decay. The learning rate is then given as

$$lr = lr_{\text{base}} \cdot min \left( \frac{step_i}{e_{\text{wup}}}, \frac{e - step_i}{e - e_{\text{wup}}} \right) ,$$

where $step_i$ is the current step, $e$ the total number of epochs and $e_{\text{wup}}$ the number of warm-up epochs. The warm-up takes place for $e_{\text{wup}} = min \left( 10, \frac{e}{20} \right)$ epochs. Warm-up reduces the dependency of early optimization steps that may cause difficulties in the longer run. Also, we weigh the cross-entropy loss with the class occurrences to re-balance the influence of an unbalanced dataset. The same is done in the accuracy computation where a balanced class accuracy is computed. For regularization, we apply an L1-regularization on the weights of the last linear layer. During the training process, a network may tend to overfit. To counteract this, we evaluate the model every $10^{\text{th}}$ epoch on the validation set and keep the model that yielded the best validation result. Furthermore, we set up a class mask to assign each class to the same number of prototypes. This is used and enforced by the class-specific losses (Clst and Sep). Changing the balance of the class assignment can be sensible to correct for imbalance in the dataset or focus on a specific class if its sentiment is more polysemous or generally more relevant. We show the hyperparameters used for our experiments, found by a grid search. The found hyperparameters were often very similar across models and datasets as the parameters mainly aimed at interpretability, not only performance. Note, not all hyperparameters were grid-searched simultaneously. We first searched for the best combination of $\lambda_1$ to $\lambda_5$ and searched for $\lambda_6$ separately because the interaction loss depends on interaction and does not directly impact the model performance during training. The best performing value for $\lambda_6$ was $0.5$. For the attention-based word selection we choose the number of heads to be 1 and $k = 4; k_{lim} = 10$.

## A.3 SENTENCE- VS. WORD-LEVEL LOSS

For clarification, we present the loss terms for sentence-level i/Proto-Trex networks in more detail: $\mathcal{L} :=$

$$\min_{\mathbf{P}, \mathbf{w}_g} \frac{1}{n} \sum_{i=1}^{n} \text{CE}(t_i, y_i) + \lambda_1 \text{Clst}(\mathbf{e}, \mathbf{p}) + \lambda_2 \text{Sep}(\mathbf{e}, \mathbf{p}) + \lambda_3 \text{Distr}(\mathbf{e}, \mathbf{p}) + \lambda_4 \text{Divers}(\mathbf{p}) + \lambda_5 ||\mathbf{w}_g|| ,$$

(5)

where

$$\text{Clst}(\mathbf{e}, \mathbf{p}) := -\frac{1}{n} \sum_{i=1}^{n} \min_{j : \mathbf{p}_j \in \mathbf{P}_{y_i}} \text{sim}(\mathbf{e}_i, \mathbf{p}_j) ,$$

(6)

$$\text{Sep}(\mathbf{e}, \mathbf{p}) := \frac{1}{n} \sum_{i=1}^{n} \min_{j : \mathbf{p}_j \notin \mathbf{P}_{y_i}} \text{sim}(\mathbf{e}_i, \mathbf{p}_j) ,$$

(7)

$$\text{Distr}(\mathbf{e}, \mathbf{p}) := -\frac{1}{n} \sum_{j=1}^{n} \min_{i \in [1,n]} \text{sim}(\mathbf{e}_i, \mathbf{p}_j) ,$$

(8)

---

[3]https://huggingface.co/transformers/model_doc/gpt2.html#gpt2tokenizer
[4]https://github.com/jayded/eraserbenchmark
[5]https://www.eraserbenchmark.com/
[6]https://pytorch.org/

| **1)** | $\lambda_1$ | $\lambda_2$ | $\lambda_3$ | $\lambda_4$ | $\lambda_5$ | $\lambda_6$ |
|---|---|---|---|---|---|---|
| | $\{0, 0.1, 0.2, 0.5\}$ | $\{0, 0.1, 0.2, 0.5\}$ | $\{0, 0.1, 0.2\}$ | $\{0, 0.1, 0.3\}$ | $\{0, 0.001\}$ | $\{0, 0.1, 0.5, 1.0\}$ |

| | **2)** | LM | $\lambda_1$ | $\lambda_2$ | $\lambda_3$ | $\lambda_4$ | $\lambda_5$ |
|---|---|---|---|---|---|---|---|
| Movie | | SBERT | 0.5 | 0.2 | 0.2 | 0.3 | 0.001 |
| | | CLIP | 0.5 | 0.2 | 0.2 | 0.3 | 0.001 |
| | | BERT | 0.2 | 0.2 | 0.2 | 0.3 | 0.001 |
| | | GPT-2 | 0.2 | 0.2 | 0.1 | 0.3 | 0.001 |
| | | DistilBERT | 0.2 | 0.2 | 0.1 | 0.3 | 0.001 |
| | **3)** | LM | $\lambda_1$ | $\lambda_2$ | $\lambda_3$ | $\lambda_4$ | $\lambda_5$ |
| Yelp | | SBERT | 0.5 | 0.2 | 0.1 | 0.3 | 0.001 |
| | | CLIP | 0.5 | 0.2 | 0.2 | 0.3 | 0.001 |
| | | BERT | 0.5 | 0.2 | 0.1 | 0.3 | 0.001 |
| | | GPT-2 | 0.5 | 0.2 | 0.2 | 0.3 | 0.001 |
| | | DistilBERT | 0.5 | 0.2 | 0.2 | 0.3 | 0.001 |
| | **4)** | LM | $\lambda_1$ | $\lambda_2$ | $\lambda_3$ | $\lambda_4$ | $\lambda_5$ |
| Toxicity | | SBERT | 0.2 | 0.1 | 0.1 | 0.1 | 0.001 |
| | | CLIP | 0.2 | 0.1 | 0.2 | 0.1 | 0.001 |
| | | BERT | 0.2 | 0.2 | 0.2 | 0.1 | 0.001 |
| | | GPT-2 | 0.2 | 0.2 | 0.1 | 0.1 | 0.001 |
| | | DistilBERT | 0.2 | 0.2 | 0.1 | 0.1 | 0.001 |

Table 4: 1) Search space for hyperparameter grid search. 2-4) Hyperparameter setup for Proto-Trex models on the three different datasets.

and

$$\text{Divers}(\mathbf{p}) \coloneqq \frac{1}{m} \sum_{\hat{j}=1}^{m} \min_{j:\mathbf{p}_j \notin \mathbf{P}_{y_i}} \text{sim}(\mathbf{p}_{\hat{j}}, \mathbf{p}_j) \,. \tag{9}$$

### A.4 RESULTS WITH CONFIDENCE INTERVALS

Due to a lack of space we shifted the results with confidence intervals to the Appendix which you can see in tab. 5.

### A.5 PROJECTION

We show the impact of projecting the prototypes onto their nearest neighbor in Tab. A.5. The projection is evaluated for sentence-level Proto-Trex networks. One can see the trade-off between interpretability and accuracy introduced by projection.

### A.6 PROVIDED EXPLANATIONS

Here we show the explanations that Proto-Trex provides. For each network, we use 10 prototypes, and for word-level networks, we use a prototype length of 4 tokens. First we extend the results of Tab. 2 for the word-level case, shown in Tab. 7. Then we show prototype lists for the models on different datasets. In Tab. 8 we show all prototypes for Proto-Trex based on the GPT-2 transformer and in Tab. 9 the prototypes based on the SBERT transformer. Both tables show the prototypes for the Yelp Open and MovieReview dataset. The results for the Jigsaw Toxicity dataset can be found in an external document in the codebase. ⚠ CONTENT WARNING: the content in this document can be disturbing due to highly toxic texts! These results do not represent the authors' opinion and show prototypical explanations provided solely by Proto-Trex.

Tab. 10 shows the pruned prototypes of iProto-Trex from Tab. 9(a). Pruning cuts off the words at the end of a sequence that go beyond two sentences or 15 tokens in total.

**a)**

| | Language Model | Yelp | Movie | Toxicity |
|---|---|---|---|---|
| sent.-level | SBERT | $94.92_{\pm0.01}$ | $84.56_{\pm0.91}$ | $\circ\mathbf{84.40}_{\pm0.03}$ |
| | SBERT (Proto-Trex) | $93.59_{\pm0.16}$ | $80.05_{\pm0.26}$ | $73.19_{\pm0.71}$ |
| | CLIP | $93.78_{\pm0.00}$ | $75.49_{\pm0.21}$ | $80.82_{\pm0.28}$ |
| | CLIP (Proto-Trex) | $87.16_{\pm1.56}$ | $63.52_{\pm0.66}$ | $67.75_{\pm2.10}$ |
| word-level | BERT | $89.41_{\pm2.01}$ | $61.45_{\pm1.16}$ | $76.88_{\pm1.33}$ |
| | BERT (Proto-Trex) | $92.10_{\pm0.08}$ | $75.51_{\pm0.42}$ | $79.35_{\pm1.09}$ |
| | GPT-2 | $93.78_{\pm0.41}$ | $\bullet\mathbf{87.05}_{\pm0.31}$ | $81.56_{\pm0.58}$ |
| | GPT-2 (Proto-Trex) | $\bullet\mathbf{95.32}_{\pm0.06}$ | $84.57_{\pm0.31}$ | $84.14_{\pm0.88}$ |
| | DistilBERT | $92.91_{\pm0.07}$ | $79.62_{\pm0.13}$ | $81.57_{\pm0.18}$ |
| | DistilBERT (Proto-Trex) | $92.71_{\pm0.03}$ | $78.64_{\pm0.14}$ | $83.39_{\pm0.47}$ |
| inter. | SBERT (iProto-Trex) | $93.81_{\pm0.03}$ | $80.24_{\pm0.31}$ | $73.36_{\pm0.78}$ |
| | GPT-2 (iProto-Trex) | $\circ\mathbf{95.25}_{\pm0.11}$ | $\circ\mathbf{84.80}_{\pm0.17}$ | $\bullet\mathbf{84.51}_{\pm0.93}$ |

**b)**

| Proto-Trex LM | Word-Selection | | Similarity | |
|---|---|---|---|---|
| | Conv. | Attn. | Cos. | L2 |
| BERT | $\mathbf{92.10}_{\pm0.08}$ | $89.66_{\pm0.92}$ | $\mathbf{92.10}_{\pm0.08}$ | $91.34_{\pm0.28}$ |
| GPT-2 | $\mathbf{95.32}_{\pm0.06}$ | $94.16_{\pm0.07}$ | $\mathbf{95.32}_{\pm0.06}$ | $94.99_{\pm0.09}$ |
| DistilBERT | $\mathbf{92.71}_{\pm0.07}$ | $91.09_{\pm0.51}$ | $\mathbf{92.71}_{\pm0.07}$ | $92.05_{\pm0.39}$ |
| SBERT | – | – | $\mathbf{93.59}_{\pm0.16}$ | $93.13_{\pm0.34}$ |
| CLIP | – | – | $\mathbf{87.16}_{\pm1.56}$ | $86.88_{\pm1.47}$ |

**c)**

| Proto-Trex LM | Faithfulness | | Acc. | |
|---|---|---|---|---|
| | Comp. | Suff. | before | after |
| SBERT | 0.22 | –0.08 | 80.05 | 69.66 |
| iSBERT | 0.21 | –0.08 | 80.24 | 70.76 |
| GPT-2 | 0.12 | 0.02 | 84.57 | 49.91 |
| iGPT-2 | 0.13 | 0.02 | 84.80 | 55.74 |

Table 5: Extension of Tab. 1 with respective confidence intervals.

| Language Model | Yelp | Movie | Toxicity |
|---|---|---|---|
| SBERT (Proto-Trex) | $94.13_{\pm0.15}$ | $83.23_{\pm0.05}$ | $83.11_{\pm0.19}$ |
| SBERT (Proto-Trex) with projection | $93.59_{\pm0.16}$ | $80.05_{\pm0.26}$ | $73.19_{\pm0.71}$ |
| CLIP (Proto-Trex) | $93.41_{\pm0.08}$ | $73.62_{\pm0.20}$ | $80.74_{\pm0.01}$ |
| CLIP (Proto-Trex) with projection | $87.16_{\pm1.56}$ | $63.52_{\pm0.66}$ | $67.75_{\pm2.10}$ |

Table 6: Impact of Projection on Proto-Trex networks. $\exp_i nteractionThemeanbalancedaccuracy(5runs)isgiveninpercentwithconfidenceintervals.$

We showcase all experiments of the user interaction in Tab. 11 which is an extension of Tab. 3. The extension includes (1) retraining the classification layer for the same number of epochs to provide a fairer comparison with the interaction methods and to exclude changes in accuracy simply due to training for more epochs, (2) pruning the original, (3) removing the original, (4) adding a new prototype without changing the original, (5) replacing the original with the user-chosen alternative "They offer a bad service." without using the interaction loss and (6) fine-tuning the original prototype.

In Fig. 12 we shows the full explanations with importance scores for the exemplary query in the motivation (*cf.* Fig. 1). It highlights the advantage of using prototype networks and the benefit of interactive learning.

| | Importance | Query: I really like the good food and kind service here. |
|---|---|---|
| word-level | 0.43·17.13=**7.37** | *P6:* Excellent baby back ribs. Creamed corn was terrific too . Came in the afternoon and was easy to get in and have a relaxing meal. Excellent service too!! |
| | 0.56·12.14=**6.80** | *P4:* Food was great...service was excellent! Will be eating there regularly. |
| | 0.39·12.89=**5.03** | *P10:* A great experience each time I come in . The employees are friendly and the food is awesome. |

Table 7: Provided Explanations by Proto-Trex networks. Highlighted words mark matching subsequences between query (top-row) and corresponding top three most similar prototypes (word-level). Importance scores (left, bold) are calculated with cosine similarity and classification weight.

| P1 | This places not authentic and the pho is not that good. Very small uncomfortable dining area and the service was horrible. |
|---|---|
| P2 | Great ambiance with a menu that is short and sweet . The food was delicious and I loved the ginger beer! Excellent service. Definitely recommend this place! |
| P3 | Waited over hour for food. Only one person making sushi. Left without food, they still charged for the one beer we had. Bad server + no food = horrible exp. |
| P4 | Food was great...service was excellent! Will be eating there regularly. |
| P5 | Called it. This place didn't stand a chance with terrible management, gross food and slow service. Too bad for the workers. |
| P6 | Excellent baby back ribs. Creamed corn was terrific too . Came in the afternoon and was easy to get in and have a relaxing meal. Excellent service too!! |
| P7 | Don't waste your money at this restaurant. Took my daughter to celebrate her birthday. They didn't even sing Happy Birthday. The food was horrible. |
| P8 | Great food, great service. Chicken tikka was awesome with fresh peppers. Iced tea had a surprising and refreshingly different flavor. |
| P9 | Worst customer service . No one even said hello after going in 3 separate times. I am an avid spender and have spent lots of money there in the past. Will not shop there again! |
| P10 | A great experience each time I come in . The employees are friendly and the food is awesome. |

(a) Yelp

| P1 | There's too much falseness to the second half, and what began as an intriguing look at youth fizzles into a dull, ridiculous attempt at heart-tugging. |
|---|---|
| P2 | Parts of the film feel a bit too much like an infomercial for ram dass's latest book aimed at the boomer demographic. But mostly it's a work that, with humor, warmth, and intelligence, captures a life interestingly lived. |
| P3 | Godard's ode to tackling life's wonderment is a rambling and incoherent manifesto about the vagueness of topical excess... In praise of love remains a ponderous and pretentious endeavor that's unfocused and tediously exasperating. |
| P4 | It's a lovely film with lovely performances by buy and accorsi. |
| P5 | It throws quirky characters, odd situations, and off-kilter dialogue at us, all as if to say, "Look at this! This is an interesting movie!" but the film itself is ultimately quite unengaging . |
| P6 | Like a Tarantino movie with heart, alias Betty is richly detailed, deftly executed and utterly absorbing. |
| P7 | A bland, obnoxious 88-minute infomercial for universal studios and its ancillary products. |
| P8 | The film fearlessly gets under the skin of the people involved... this makes it not only a detailed historical document, but an engaging and moving portrait of a subculture. |
| P9 | A sad and rote exercise in milking a played-out idea – a straight guy has to dress up in drag – that shockingly manages to be even worse than its title would imply. |
| P10 | an inventive, absorbing movie that's as hard to classify as it is hard to resist. |

(b) Movie

Table 8: Proto-Trex GPT-2 Prototypes. The prototypes are received with GPT-2 for the Yelp Open (a) and MovieReview (b) dataset. The prototypical subsequences provided by GPT-2 are highlighted in color.

| P1 | This place is horrible. The staff is rude and totally incompetent. Jose was horrible and is a poor excuse for a customer representative. |
| P2 | This place is really high quality and the service is amazing and awesome! Jayden was very helpful and was prompt and attentive. Will come back as the quality is so good and the service made it! |
| P3 | Terrible delivery service. People are mean and don't care about their customers service. I will not ever come back to this place. Also the food is small, not very good and too expensive. |
| P4 | Incredibly delicious!!! Service was great and food was awesome. Will definitely come back! |
| P5 | They sat us 30 minutes late for our reservation and didn't get our entree for 1.5 hours after being seated. The service was terrible. |
| P6 | I found this place very inviting and welcoming. The food is great so as the servers.. love the food and the service is phenomenal!! Well done everyone..! |
| P7 | Horrible customer service. Not helpful at all and very rude. Very disappointed and will not go back to this location. |
| P8 | Oliver Rocks! Great hidden gem in the middle of Mandalay Bay! Great friends great times |
| P9 | They literally treat you like you are bothering them. No customer service skills. Substandard work. |
| P10 | Fun and friendly atmosphere, fantastic selection. The sushi is so fresh and the flavors are WOW. |

(a) Yelp

| P1 | ...in the pile of useless actioners from mtv schmucks who don't know how to tell a story for more than four minutes. |
| P2 | the solid filmmaking and convincing characters makes this a high water mark for this genre. |
| P3 | dull, lifeless, and amateurishly assembled. |
| P4 | it's a wise and powerful tale of race and culture forcefully told, with superb performances throughout. |
| P5 | stale, futile scenario. |
| P6 | a real winner – smart, funny, subtle, and resonant. |
| P7 | plodding, poorly written, murky and weakly acted, the picture feels as if everyone making it lost their movie mojo. |
| P8 | an enthralling, entertaining feature. |
| P9 | fails in making this character understandable, in getting under her skin, in exploring motivation... well before the end, the film grows as dull as its characters, about whose fate it is hard to care. |
| P10 | this delicately observed story, deeply felt and masterfully stylized, is a triumph for its maverick director. |

(b) Movie

Table 9: Proto-Trex SBERT Prototypes. The prototypes are received with SBERT for the Yelp Open (a) and MovieReview (b) dataset.

| P1 | This place is horrible. The staff is rude and totally incompetent. Jose was horrible and is |
| P2 | This place is really high quality and the service is amazing and awesome! Jayden was |
| P3 | Terrible delivery service. People are mean and don't care about their customers service. I will |
| P4 | Incredibly delicious!!! Service was great and food was awesome. Will definitely come back! |
| P5 | They sat us 30 minutes late for our reservation and didn't get our entree for |
| P6 | I found this place very inviting and welcoming. The food is great so as the servers.. |
| P7 | Horrible customer service. Not helpful at all and very rude. Very disappointed and will not go |
| P8 | Oliver Rocks! Great hidden gem in the middle of Mandalay Bay! |
| P9 | They literally treat you like you are bothering them. No customer service skills. Substandard work. |
| P10 | Fun and friendly atmosphere, fantastic selection. The sushi is so fresh and the flavors are |

Table 10: Pruning Prototypes with iProto-Trex. It shows the prototypes from Tab. 9(a) after pruning has been applied. Pruning reduces the sequences in length while preserving the sentiment. The accuracy remains the same (*cf.* Tab. 11). Prototypes 4 and 9 are not pruned as this would alter the sentiment too much.

| Method | Acc. | Prototype |
|---|---|---|
| no interaction | 93.64 | Horrible customer service and service does not care about safety features. That's all I'm going to say. Oh they also don't care about their customers |
| retrain | 93.64 | Horrible customer service and service does not care about safety features. That's all I'm going to say. Oh they also don't care about their customers |
| prune | 93.64 | Horrible customer service and service does not care about safety features. That's all I 'm |
| remove | 93.61 | – |
| add | 93.79 | They offer a bad service. |
| replace | 93.78 | They offer a bad service. |
| re-initialize | 93.80 | Terrible delivery service. People are mean and don't care about their customers service. I will not ever come back to this place. Also the food is small, not very good and too expensive. |
| fine-tune | 93.81 | Terrible delivery service. People are mean and don't care about their customers service. I will not ever come back to this place. Also the food is small, not very good and too expensive. |
| soft replace (0.5) | 93.81 | Terrible delivery service. People are mean and don't care about their customers service. I will not ever come back to this place. Also the food is small, not very good and too expensive. |
| soft replace (0.9) | 93.79 | I really don't recommend this place. The food is not good, service is bad. The entertainment is so cheesy. Not good |
| soft replace (1.0) | 93.79 | They offer a bad service. |

Table 11: User Interaction with iProto-Trex. Each row shows a different interaction method with the balanced accuracy on the test set conducted on Yelp Open dataset. The interaction methods are able to remove the unwanted prototype while incorporating more knowledge and hence more interpretability along with a higher accuracy.

| | Importance | Query: Staff is available to assist, but limited to the knowledge and understanding of the individual |
|---|---|---|
| **Proto-Trex** | $0.58 \cdot 6.95 = \mathbf{4.01}$ | They literally treat you like you are bothering them. No customer service skills. Substandard work. |
| | $0.38 \cdot 2.61 = \mathbf{1.00}$ | They sat us 30 minutes late for our reservation and didn't get our entree for 1.5 hours after being seated. The service was terrible. |
| | $0.36 \cdot 2.02 = \mathbf{0.73}$ | This place is horrible. The staff is rude and totally incompetent. Jose was horrible and is a poor excuse for a customer representative. |
| **iProto-Trex** | $0.50 \cdot 3.25 = \mathbf{1.63}$ | They offer a bad service. |
| | $0.38 \cdot 4.17 = \mathbf{1.58}$ | They sat us 30 minutes late for our reservation and didn't get our entree for 1.5 hours after being seated. The service was terrible. |
| | $0.36 \cdot 3.01 = \mathbf{1.08}$ | This place is horrible. The staff is rude and totally incompetent. Jose was horrible and is a poor excuse for a customer representative. |

Table 12: Top-3 Explanations for Query in the Introduction (Fig. 1). The first three rows depict explanations with sentence-level Proto-Trex with the SBERT transformer, while the last three show the top three explanations iProto-Trex. On the left side, one can see the importance scores for the classification.