# OpenReview forum: "Interactively Generating Explanations for Transformer Language Models"
_ICLR.cc/2022/Conference — ICLR 2022 Submitted_

### Official Review · Reviewer_hWyb · 2021-11-01

**Correctness:** 3
**Technical Novelty And Significance:** 3
**Empirical Novelty And Significance:** 3
**Recommendation:** 5
**Confidence:** 4

**Main Review:**

Strengths: The paper touches upon several interesting threads: interpretability, case based reasoning and using user interactions for improving models. It is well written, with a comprehensive set of experiments.

Weaknesses:
- Hyperparameters: My initial impression of the method is that it involves far too many hyperparameters (various $\lambda$ for combining losses, choosing similarity metrics etc). How hard was tuning hyperparameters?
- Results:
    - different models respond differently to the proposed approach. For instance, we see large performance gains on BERT but performance drops for SBERT. It would be good to know why this happens.
    - User interaction results are not convincing. From Table-3, it seems like the interactions barely help, even though the prototype changes sometimes. Could the authors comment more about this?
    - Can the authors provide with confidence intervals for all results so we can compare across different settings better?
Prototype quality: prototypes do not look very convincing. I glanced at the prototypes in the Appendix and they seem to be quite redundant / of low diversity. For instance, from Table-6 a lot of the prototypes either model “bad service and bad food” or “good service and good food” (P2/P4/P6/P8/P10). However yelp also has examples with “bad service, good food”, “good food, bad service”, examples not related to restaurants at all but the prototypes do not uncover these.
Faithfulness: How competitive are these numbers with post-hoc methods? I think for presentation purposes, it would be good to also compare this with SHAP/LIME/IG.

Overall, I think this paper is currently borderline. However, I’m happy to increase my score if the weaknesses can be quickly addressed —- the main ones are providing evidence that the learnt prototypes can be diverse and cover a lot more of the “modes” in the training distribution, providing a more comprehensive comparison with posthoc interpretability approaches, and confidence intervals.


**Summary Of The Paper:**

This paper introduces a method for improving interpretability of black box transformer based text classifiers. The approach is based on “case based reasoning” where the network classifies an input by comparing it against a library of learnt prototypes (each with a corresponding label) and classifying the input based on a weighted similarity with the prototypes. Thus, in a sense it is more interpretable than an end-to-end classifier since a user can directly look at the similarity scores with the prototypes to understand the labeling decision. The approach also promises to be more “faithful” than post-hoc interpretability methods since the predictions are based directly on similarity scores. The paper also shows that end-users of the system can interact with it by either editing prototypes (if they are experts) or providing weak feedback about which prototypes are good, and this feedback can be integrated to re-learn better prototypes. From experiments we see that their approach is competitive with standard end-to-end finetuning while being more interpretable. Based on sufficiency and comprehensiveness scores, we see some evidence that the explanations from prototypes are faithful, though it is unclear if these scores are competitive.


**Summary Of The Review:**

While i think this work combines many interesting threads in interpretability, using user feedback and case based reasoning, the paper in its current state is not ready. I’m happy to increase my score / recommendation if the weaknesses are addressed, however.

---

> ### Author Response · Authors · 2021-11-16
> **The commendation on the significance of our work and the well presented approach is appreciated. Based on the feedback on unclear parts we conducted additional experiments to improve our work.**
>
> We appreciate that our approach is valued and are grateful for the feedback. While the reviewer has commended our experiments, we understand that the limited space in the main body led to open questions targeted in the following:
>
> **Many hyperparameters - how hard was tuning?** \
> Thanks for hinting at this issue; we missed putting the final hyperparameters in our work and added them to Appendix A.1. The training time depends very much on the dataset and model and took between 1 min (sentence-level and smallest dataset) and 3 hours (word-level and largest dataset) on one of our GPUs (available: 16 Nvidia Tesla V100). As written in Appendix A.1, we applied grid-search for hyper-parameter search. All in all, it took us roughly two days of computation time, which we regard as reasonably efficient. Prior work on prototype networks (Li et al., 2018; Chen et al., 2019; Hase et al., 2019; Ming et al., 2019) gave us intuition on sensible hyperparameter ranges, which helped decrease search space and thus time. To give readers better insights, we added the hyperparameter list to Appendix A.1. In addition, one can see that the hyperparameters were mostly similar across datasets and models, indicating that it does not result in enormous time consumption for tuning.
>
> **Different models respond differently; why?** \
> The difference in performance is mainly due to projection. We explain this in greater detail in Section 3.5, Section 6 in paragraph "(Q1,2) Trade-off accuracy and interpretability" and Appendix A.4. The projection "[...] has the advantage of being decoded precisely on the spot and providing more certainty for the explanatory power. The disadvantage, however, is that a prototype may be located sub-optimally, which could lead to performance losses". The projection is used for sentence-level models (like SBERT) why they have greater drops in accuracy than the word-level models (like BERT). We also found a typo in the accuracy for BERT on the MovieReview dataset, which we corrected.
>
> **Unclear user interaction results (Tab.3)** \
> [Misunderstanding]: In Tab.3, we aim to increase the performance in terms of interpretability, not accuracy. We use Tab.3 to answer Q2 on "Interactive Learning", where we use the human in the loop to showcase possible interaction scenarios and their potential for interpretability. The accuracy scores only show that interaction is not affecting the network's performance. Tab.3 shall show that users can interact with the network, regardless of whether they are experts (weak knowledge with soft interaction), and adjust the network to their personal preferences. This can be very helpful for trust and interpretability. We adapted the table's caption and reduced the font size of the accuracy scores to make this clearer and prevent misunderstandings.
>
> **Confidence intervals** \
> Thanks for your hint; due to space reasons, we preliminary added the missing confidence intervals to Appendix A.4. For the camera-ready version, we consider moving it to the main text.
>
> **Semantic diversity of prototypes** \
> Our current setup is not encouraging the network to learn prototypes close to the decision boundary, i.e. prototypes containing a positive and a negative and, thus, an overall ambiguous sentiment. If we increase the number of prototypes, e.g. $m=100$, it is more likely to obtain prototypes closer to the decision boundary due to the diversity and distribution loss. The experiment with more prototypes yielded, for example: "Cute atmosphere. But the product isn't drinkable. [...]". Also, prototypes related to doctors, clothes and car shops are present if the number of prototypes increases for the Yelp Open dataset. So, in general, it is possible to obtain more diverse prototypes, but there is a trade-off: a large number of prototypes reduces the interpretability and maybe provides ambiguous prototypical explanations. We focus on learning prototypes of a clear sentiment which may be more helpful for the user in understanding the decision respectively explanation. Together with the code, we add the results in the supplementary material for reproducibility and further insights.
>
> **Faithfulness: How competitive are these numbers with post-hoc methods?** \
> We agree that it is sensible to consider this comparison for presentation purposes. Related work (e.g. DeYoung et al., 2020) also gives insight into the faithfulness of these approaches; although they use different models, the faithfulness scores are not on par with ours. One has to carefully distinguish 1) the faithfulness of a model (does the model use the "right" input parts?) and 2) the faithfulness of the explanation (does the explanation truly explain the classification?). It is more challenging to evaluate the faithfulness of post-hoc explanations as 2) removing parts of the explanation goes hand in hand with 1) removing parts of the input. We currently evaluate the post hoc methods on faithfulness and keep you posted about the results.

---

> > ### Author Response · Authors · 2021-11-19
> > **Update Faithfulness**
> >
> > We computed the accuracy for the SBERT and GPT-2 baselines, where we removed the top 10% most influential words based on the LIME importance scores. The accuracy drops for SBERT from 84 to 63% and GPT-2 from 87 to 72%. So, we can see that LIME, to some extent, also seems to faithfully explain via important parts, while prototype networks, especially for word-level LMs, seem to be much more faithful. We will include the values in Tab.1c and our discussion on faithful classifiers/ explanations.

---

### Official Review · Reviewer_QkZJ · 2021-11-02

**Correctness:** 3
**Technical Novelty And Significance:** 2
**Empirical Novelty And Significance:** 2
**Recommendation:** 5
**Confidence:** 3

**Main Review:**

Strengths:

1. The proposed Proto-Trex/iProto-Trex are technically sound.
2. Through several example cases, the proposed model could generate reasonable explanations.


Weaknesses:

1. The training loss of the model seems to be too complicated. In Eq.1, there are 6 parts for the overall training loss, and each of them associates with a lambda term (I checked the Appendix, but failed to find how you set lambda terms). Each of them sounds reasonable, but unfortunately, there is no specific ablations on how these parts attribute to the final prediction/explanation.
2. According to the illustrations at the end of Section 3.4, there are many hyper-parameters should be set (along with lambda terms in training loss). How to determine these hyperparameters? If we move to a new dataset, we might lost in tuning these hyperparams.
3. As the model incorporates additional modules (compared to pure classification model), it is questionable how efficient is Proto-Trex (both for parameter size and inference time).


Comments:

1. It is unclear where are the results from in Table 1b and 1c. Yelp? Movie? Toxicity?


**Summary Of The Paper:**

This paper proposes Proto-Trex model to increase the interpretability of the text classification systems. The proposed model mainly adds a bunch of prototype layers to learn the similarity between the query and prototypes. They also propose an extension caleld iProto-Trex to interactively learn from users' feedbacks. Experimental results and example cases show that the Proto-Trex could give comparable classification accuracy compared to plain classification model (w/o explanation) and could provide reasonable explanations.

**Summary Of The Review:**

The paper addresses an important issue in explainable natural language processing models. The proposed model is reasonable, and the results seem to be OK. However, many issues are left unclear, such as its efficiency, hyper-parameter tuning, etc. I tend to lean towards weak rejection at the moment.

---

> ### Author Response · Authors · 2021-11-15
> **The emphasis on technical soundness and a solid experimental evaluation is appreciated. Based on the feedback on unclear parts, we propose to include additional explanations in the appendix. Also, further clarifications are provided to the reviewer's questions.**
>
> We thank you for the time and effort you put into the review. We are delighted that you liked our submission, our proposed method's technical soundness, and the experimental results, i.e. the generated prototypical explanations. With respect to the concerns that the reviewer is not 100% sure about, we are happy to provide additional clarifications below to the questions you raised:
>
> **1. The loss is too complicated, and the hyperparameters are unclear.** \
> Thank you for hinting at this issue; as we missed putting the final hyperparameters in our work, we added them to Appendix A.1. The training time depends very much on the dataset and model, and as written in Appendix A.1, we applied grid-search for hyper-parameter search. It took us roughly two days of computation time on our GPUs (in total available: 16 Nvidia Tesla V100), which we regard as reasonably efficient (see also reply bullet point 1 to Reviewer 4: hWyb). Prior work on prototype networks (Li et al., 2018; Chen et al., 2019, Hase et al., 2019; Ming et al., 2019) gave us intuition on sensible hyperparameter ranges, which helped decrease search space and thus time. As we incorporated $\lambda_i=0$ in our grid search, we also checked the necessity of each loss term. Besides that, it is crucial to remember that the additional loss terms are not used solely for optimizing accuracy but for interpretability.
>
> **2. How to determine the hyperparameters in Section 3.4? We might get lost in tuning those.** \
> We agree that an increasing number of hyperparameters affects the time consumption for optimizing a model. However, we also see the benefit of using more sophisticated methods to improve performance which is a general trade-off. In Section 3.4. we evaluated the hyperparameters within each module by hand as it nearly did not affect the performance but the interpretability. The optimization of the word-selection hyperparameters is mainly human rather than data specific. "How many words are needed to have a good prototypical explanation?" is a question for future research, e.g. a user study. Furthermore, the hyperparameters (especially those in 3.4.) are similar even across different datasets, which shows that it may already be sufficient to use them across datasets.
>
> **3. How efficient is the model in terms of size and inference time?** \
> The parameter size of the baseline models and the Proto-Trex models is precisely the same as the additional modules do not have any parameters. For sentence-level models, the inference time is comparable with the baseline, as we have $e * w_k$ computations for the baseline and $(e*p) + w_l$ for Proto-Trex with $k>>l$. For the word-level models, we have a slightly increased inference time due to the word-selection module. If you keep the required number of computations of a single forward pass in a large-scale transformer in mind, it becomes more apparent that the increase in computation is relatively negligible.
>
> **The used dataset for Tab.1b/c is unclear.** \
> In the text, we mention, e.g. in Q3 in Section 6, that the results in c) come from the MovieReview and in b) from the Yelp dataset. We agree that the clarity in the paper benefits from putting more emphasis on the datasets, so we added the datasets to the table's caption.

---

### Official Review · Reviewer_cN7n · 2021-11-02

**Correctness:** 2
**Technical Novelty And Significance:** 3
**Empirical Novelty And Significance:** 2
**Recommendation:** 5
**Confidence:** 4

**Main Review:**

This paper seems motivated by a prior NeurIPS 19 work "This Looks Like That..." in the sense that the architecture and loss designs are sourced from there. The nice thing about this paper is that it focuses on NLP tasks, so the framework could potentially benefit the explanation community. The early part of this paper is very straightforward and intuitive. Related works have limited coverage. The explanation generation part is vague. The experiment section is weak. Analysis on generated explanation is also weak.

The architecture relies on using prototypes which are nicely discussed in this paper. The problem if improving interpretability without trading off downstream task F1 is interesting since the trade-off was common among prior works. However a couple confusing points still.

1. There are many lines of explanation works, such as those use prompt engineering, information bottleneck, and purely generative approaches. This paper has limited coverage on these topics. Touching different approaches is important here since the way ProtoTrex handle explanation might not easily extend to all other cases.

2. Even though this design was from the NeurIPS 19 paper, but in the task of NLP, how does the prototype embeddings compare against the label-wise weights in the final classification layer? This is to imagine that, without the use of the complicated loss in Eq 1, how does simply treating the label-wise embeddings as prototypes perform?

3. The explanation generation (sec 3.5) needs elaboration. It seems this paper uses prototype embedding to find a training example as it nearest neighbor, and then use this data point as explanation to its prediction. This design has certain limitations: a) not context/example dependent; b) this is hardly generation, instead, it is more in line with salience-based explanation works. Tab2 indeed shows some examples with explanation that partially depends on the input example. But no idea how they were generated.

4. Tab 1b is confusing. I don't understand what each number means. This paper very briefly went over some statements without getting into details.

5. I don't see a solid explanation evaluation in this paper. Tab 1c shows rationale performances however these numbers are quite low compared with prior works (e.g. Paranjape's work at EMNLP 20). And not sure if rationale performances are based on token or sentence selection. Either way, this evaluation has nothing to do with generation. And when it comes to generative explanations, ideally, there should be some human-based evaluation over a subset of testing data. But no such thing in this paper.

**Summary Of The Paper:**

This paper aims to model explanation and task prediction such that task performances are not (or less) traded off for interpretability. It proposes a novel framework for transformer models where classification and explanation generation are based on shared prototype embeddings which are learnt from training data by a combination of losses. The framework is also compatible with settings that requires human in the loop for extra supervision on prototype learning. Experiment results show that adding the proposed ProtoTrex benefit task performances on 3 sentiment classification tasks.

**Summary Of The Review:**

I think the architecture has novelty when it comes to NLP tasks. And this work could benefit the explanation community. However, I found the experiment results are confusing. To the best degree, it offers marginal improvement over the best baselines in terms of task F1. When it comes performance of explanation, I only see confusing numbers, thus no conclusion can be made. Analysis on generated explanation is another weak point since it's absent.

---

> ### Author Response · Authors · 2021-11-15
> **The positive evaluation on relevance of our work for the NLP community is appreciated. We suggest further improvements based on the valuable feedback**
>
> We thank the reviewer for the valuable feedback and are glad to see the reviewer agree on the relevance for the XAI community integrating XAI methods into the field of NLP.
>
> **1. Related work is limited and needs an extension.** \
> [Misunderstanding]: Thanks for referring us to further related work. We agree that information extraction through bottlenecks, generative approaches (learning to explain), and prompt engineering are exciting and emerging topics in the field of explainable AI. However, all three tackle different problems, which we do not consider close enough, so we did not regard it as related work.
>
> **2. Why do we need multiple prototypes per class, making it necessary to use a complex loss?** \
> If we use only two prototypes without a weighting, i.e. both have equal weights for the respective class, we achieve about 5-10% lower accuracy but can still classify pretty well. More prototypes enable more diverse prototypical explanations, such that, e.g. positive explanations are not always related to just the same prototype. This can help to improve interpretability. The original paper ("This looks like that") uses multiple prototypes to account for different prototypical parts of a bird, e.g. the shape of the head, wings, tails, or their color. For achieving a decent accuracy, it may already suffice to use only one prototype per class. However, this also potentially leads to bad interpretability as we do not classify based on close (similar) cases but farther away (less similar) cases.
>
> **3. Lack of clarity on the term explanation "generation".** \
> In the context of (auto-regressive) transformer LM the word "generating" may be misleading; instead, we prefer "providing". Our proposed method provides a learned set of training examples as explanations. This means they are "generated" from the training data, which we regard as a constrained generation form. Furthermore, the resulting set of explanations/contexts is consequently limited by the number of prototypes set beforehand. Unlike post-hoc explanations, we do not get an individual explanation for a query, which is fundamental in the method design. Still, each query is explained by the nearest prototypical examples. It may not be necessary to have an infinite number of possible explanations, especially if we want to classify only two classes.
>
> **4. Tab.1b) is unclear.** \
> We fixed the caption in the table to improve the clarity and refer to the text in Section 6 paragraph ''(Q1) Ablation study.'', where we reference and describe the table in more detail; more precisely, we explain the conducted ablation study and explain our prototype module choices.
>
> **5.1 The evaluation on faithfulness is weak and misses the F1 score.** \
> [Misunderstanding]: We think there is a misunderstanding and like to clarify it. Your proposed related paper from Paranjape et al. uses the IOU and Token F1 score from DeYoung et al. (EraserBenchmark) to benchmark their experiments on the "agreement with human rationales", i.e. the performance of rationale extraction. Since we want to measure faithfulness, we use different metrics from that paper, i.e. the comprehensiveness (Comp) and sufficiency (Suff) score; hence the values of your paper are not comparable (please compare Section 4.1 and 4.2 in DeYoung et al.: https://arxiv.org/pdf/1911.03429.pdf). DeYoung uses the Comp and Suff score to evaluate the faithfulness on the MovieReview dataset. We strictly followed their approach to benchmark our model. Comp and Suff are common choices and state-of-the-art to evaluate faithfulness. \
> Furthermore, to account for different aspects of faithfulness, we 1) removed the rationale in the input data to check whether the model is still certain about the classification (DeYoung's approach), we showed that there is a significant loss in confidence about the class prediction; and we additionally 2) removed the prototype that is most influential for the prediction to check how influential (faithful) the prototype really was. As the accuracy decreased to a large extent, the influence on the prediction was huge, i.e. the prototype was really used for the classification and served as a faithful explanation. Both experiments emphasize that 1) prototypes relate to the rationale in the input, i.e. need the rationale to correctly and confidently classify the query and that 2) the provided prototypical explanation is actually used for the classification.
>
> **5.2 The explanations lack a human-based evaluation** \
> This submission aims to establish prototype networks in the NLP community, especially for transformer language models. A user study is a promising future venue to confirm and further investigate this approach's explainability for black-box models; we have yet been unable to do one. We agree that our paper can benefit from a user study, albeit related work on prototypical networks has already shown improved interpretability (e.g. Hase et al., 2020, "Evaluating Explainable AI").

---

### Official Review · Reviewer_yfzh · 2021-11-08

**Correctness:** 3
**Technical Novelty And Significance:** 2
**Empirical Novelty And Significance:** 2
**Recommendation:** 3
**Confidence:** 3

**Main Review:**

Strengths
- S1 - The idea of choosing a similar example from the train data as an explanation is an interesting idea.

Weaknesses and Questions:
- W1 - I'm quite unsure about the set of prototypes. It seems not described in the main contents. I checked examples of prototypes in the appendix but the 1:1 label distribution (positive vs. negative) seems to me they are human-picked.
- W1-Q1 - If the set of prototypes pre-defined, do we really need to train the prototype? what if we just compute sentence similarity between input and the prototype?
- W2 - The main table is not clear, especially on faithfulness. What if we just consider the random prototype as a prototype? what is the faithfulness of that random prototype then?
- Q2 - Just curious, in the introduction, why the post-hoc explanation is potentially reporting-bias? Isn't this method also potentially reporting-bias since it only uses samples from train data as prototypes?

**Summary Of The Paper:**

The premise of the paper is that example train instances with the corresponding label can be an effective explanation of the model's prediction rather than post-hoc explanation which is not deterministic. The framework jointly trains label prediction and prototype clustering. Then, the framework computes the similarity between input and prototypes to retrieve the most similar prototype as an explanation while predicting the label as well. Moreover, the framework can reflect human-in-the-loop feedback on the prototype.
The primary result of the paper is that their framework can show the proper explanation by a similar example while preserving the performance.

**Summary Of The Review:**

My overall disposition towards the paper is indifferent although the paper proposes an interesting idea. It is a bit difficult to follow the method due to a lack of information. More details are needed especially on experimental settings, and methods.

---

> ### Author Response · Authors · 2021-11-15
> **The positive assessment of our idea is appreciated. We reply to the remaining concerns, erase potential misunderstandings and propose using the constructive feedback in our revised version.**
>
> We thank you for your time and comments. First of all, we are encouraged that the reviewer liked our idea of using prototypical examples (S1) to improve the explainability for decisions of black-box models.  Besides valid inquires, unfortunately, there seem to be several misunderstandings, which we like to address in the following. We will revise our manuscript accordingly to clarify the potential misunderstanding you pointed to.
>
> W1: **The set of prototypes is not sufficiently described and seems to be human-picked.** \
> [Misunderstanding]: The set of prototypes and the label distribution are clearly described in paragraph "Architecture" in Section 3.1 and Appendix A.2. The label distribution is a design choice made by the Proto-Trex creator, but the prototypes are trained end-to-end together with the model's parameters. We use a class mask with a 1:1 label distribution to enforce the class-specific prototype loss, resulting in prototypes of altering polarity. The class mask distribution can be changed according to the users' preferences or dataset imbalance.
>
> W1-Q1: **The set of prototypes is pre-defined; why do we need to train them, and can't you simply compute sentence similarities?** \
> [Misunderstanding]: The set of prototypes is not pre-defined. In fact, the prototype vectors are randomly initialized; Section 3.1. explains that they are a "trainable parameter [...] learned during the training process". The random initialization of the prototypes is not mentioned in the current main text. To prevent further misunderstanding, we add this to the experimental section. \
> Prototypical networks classify a sample based on its similarity to each prototype. The (sole) purpose of the above-mentioned mask (see reply to W1) is in the loss term to ensure that we use prototypes of the same class, and therefore avoid negative reasoning(see Section 3.2.). The computed similarity is forwarded to the classification layer, which weights the similarities to infer the class label. We use the ground truth label only to compute the Cross-Entropy Loss. \
> However, the prototype initialization is a design choice, and related work also initialized randomly or used beforehand clustering. Besides the random initialization, we also conducted experiments where we initialized the prototypes ourselves. Nevertheless, we discarded this approach to prevent unnecessary bias as this did not significantly improve the performance.
>
> W2: **What if we just consider a model with random prototypes for the faithfulness computation?** \
> That is an interesting question: If the prototypes are randomly initialized and not further learned during training, i.e. only the parameters of the linear classification layer are updated, we obtain models with accuracy scores between 55 and 65% on the respective test set. As we regarded it as not meaningful to compute the faithfulness of a bad performing model, we omitted this experiment.
>
> Q2: **Why are prototype networks suffering less from reporting bias than post-hoc explainers?** \
> We understand the source of this concern and like to explain it here in greater detail. Post-hoc explainers and interpretability methods show the model's reasoning only for a specific sample that the author selects. Such a sample-based post-hoc method may encourage overinterpreting the influence of a word or concept for the classification in one sample, where the same word or concept would not be a meaningful influence in another sample (leading to reporting bias). We are, in contrast, motivated to show all learned prototypes of a model: prototypes are a substantial part of the model's reasoning process for every sample. At the same time, we do not claim to eliminate reporting bias like this; identifying all prototypical explanations in prototype networks grants users more insights into the reasoning process.

---

### Author Response · Authors · 2021-11-25
**Checking in ...**

We thank the reviewers for their diligent reviewing and comments. We have provided a complete set of responses to their comments and updated the paper based on the reviewers’ feedback. We would be happy to receive feedback on our responses before the discussion phase ends.

---

### Decision · Program_Chairs · 2022-01-20

**Decision:**

Reject

**Comment:**

The authors propose a method—"Proto-Trex"—that incorporates prototype networks into text classification architectures to facilitate model explanations via presentation of similar training examples. There was agreement that the direction here is promising and the work contains some nice ideas and a good initial set of evaluations. However, the presentation can be improved substantially to better situate the contribution with respect to related work (and clarify the specific contributions on offer here), and to clarify the key technical details of the proposed approach.